# Models of Support for Caregivers and Patients with the Post-COVID-19 Condition: A Scoping Review

**DOI:** 10.3390/ijerph20032563

**Published:** 2023-01-31

**Authors:** Tahissa Frota Cavalcante, Caroline Evaristo Lourenço, José Erivelton de Souza Maciel Ferreira, Lídia Rocha Oliveira, João Cruz Neto, Josemberg Pereira Amaro, Rafaella Pessoa Moreira

**Affiliations:** Health Sciences Institute, University for International Integration of the Afro-Brazilian Lusophony (UNILAB), Redençao 62790-000, Ceará, Brazil

**Keywords:** post-COVID-19 condition, caregivers, models of support, models of care, public health

## Abstract

Background: In December 2019, an outbreak of the coronavirus disease 2019 (COVID-19), caused by the novel severe acute respiratory syndrome coronavirus 2 (SARS-CoV-2), occurred in the city of Wuhan, China. On 30 January 2020, the World Health Organization declared the outbreak a public health emergency of international concern. In October 2021, with the advancement of the disease, the World Health Organization defined the post-COVID-19 condition. The post-COVID-19 condition occurs in individuals with a history of probable or confirmed infection with SARS-CoV-2, usually 3 months after the onset of the disease. The chronicity of COVID-19 has increased the importance of recognizing caregivers and their needs. Methods: We conducted a scoping review following international guidelines to map the models of support for caregivers and patients with the post-COVID-19 condition. The searches were conducted in electronic databases and the grey literature. The Population, Concept, and Context framework was used: Population: patients with the post-COVID-19 condition and caregivers; Concept: models of caregiver and patient support; and Context: post-COVID-19 condition. A total of 3258 records were identified through the electronic search, and 20 articles were included in the final sample. Results: The studies approached existing guidelines and health policies for post-COVID-19 condition patients and support services for patients and home caregivers such as telerehabilitation, multidisciplinary care, hybrid models of care, and follow-up services. Only one study specifically addressed the home caregivers of patients with this clinical condition. Conclusions: The review indicates that strategies such as telerehabilitation are effective for training and monitoring the patient–family dyad, but the conditions of access and digital literacy must be considered.

## 1. Introduction

On 31 December 2019, China alerted the World Health Organization (WHO) about several cases of pneumonia of unknown cause in the city of Wuhan, Hubei Province, central China. On 7 January, a novel coronavirus, originally abbreviated as 2019-nCoV by the WHO, was identified from a patient’s swab sample [1]. This pathogen was later renamed severe acute respiratory syndrome coronavirus 2 (SARS-CoV-2) by the Coronavirus Study Group [2], and the disease was named coronavirus disease 2019 (COVID-19). As of 30 January, 7736 confirmed cases and 12,167 suspected cases had been reported in China, and 82 confirmed cases were detected in 18 other countries. The same day, the WHO declared the SARS-CoV-2 outbreak a public health emergency of international concern (PHEIC) [3].

Clinical manifestations of 2019-nCoV infection include fever, dry cough, dyspnea, chest pain, fatigue, and myalgia. Less common symptoms include headache, dizziness, abdominal pain, diarrhea, nausea, and vomiting [4]. The sequelae resulting from the acute phase of COVID-19 have been reported in the literature [1,2,3,4,5]. The post-COVID-19 condition is an illness that occurs in individuals with a history of probable or confirmed infection with SARS-CoV-2, usually 3 months after the onset of COVID-19, with symptoms that last at least 2 months and cannot be explained by an alternative diagnosis. Common symptoms include fatigue, shortness of breath, and cognitive dysfunction, which often impact daily functioning. Symptoms can be a new onset after initial recovery from an acute episode of COVID-19 or persist from the initial illness, as well as fluctuate or relapse over time [1,2].

A recent cohort study of 1191 patients who were hospitalized with COVID-19 and discharged from Wuhan, China, with 6-month, 12-month, and 2-year analyses, demonstrated that 6-month symptoms compared to 2-year symptoms reduced but are still considered for concern, as dyspnea persists in 14% of patients and anxiety or depression in 12% of patients after 2 years of infection [1,2]. Two years after the acute phase of the disease, survivors have a lower health status than those who did not have COVID-19, matched by sex, age, and comorbidities [2].

Systemic inflammation triggered by SARS-CoV-2 virus infection may further contribute to neuroinflammatory processes and increase susceptibility to neurological or thrombotic syndromes. Central Nervous System (CNS) infections can promote the development of neurodegenerative diseases in at-risk individuals. Thus, the impact of SARS-CoV-2 on the CNS can lead to direct neurological changes, worsen pre-existing neurological conditions/increase susceptibility, or aggravate the damage caused by other events, promoting cerebrovascular and cardiovascular diseases [6].

Such facts have drawn attention to the chronicity of COVID-19, providing survivors with sequelae and functional disabilities that require long-term intensive rehabilitation and home care [2,7]. It is also noteworthy that the partnership between rehabilitation/home care professionals and caregivers is crucial to guarantee support to people who suddenly assume the role of caregiver and are not prepared for it [8].

The caregiver is the link between the healthcare team and the patient. It is the person with whom healthcare professionals share responsibilities. In many cases, the caregiver is seen and treated as a simple performer of procedures and obliged to comply with what was unilaterally prescribed by the home care professional. This situation produces tension, which can negatively impact the relationships established in the households and the quality of care provided by both the caregiver and health professionals [9,10].

The COVID-19 pandemic brought, in addition to the chronicity of COVID-19, demands for new roles within the family context, including an additional role for family members—that of the caregiver. This context is added to the most stressful care activities such as the fear of new contamination, loss of a loved one, accumulation of multiple functions, social isolation, and reduced economic and social support for families. Therefore, the following question has emerged: Which models of support for caregivers and patients with the post-COVID-19 condition exist?

It is worth noting that there are no scoping reviews on the proposed theme registered in the OSF Registries platform. The present review is important because it aims to fill knowledge gaps. This review will fill the knowledge gap about the existing support models for caregivers of people with the post-COVID-19 condition, as these data are not yet available. It will synthesize models of home care support for caregivers and patients that promote and recover these persons’ health. Furthermore, it will present important care models for implementation by managers and provide new perspectives for research. Given the context above, this scoping review aimed to map the models of support for caregivers and patients with the post-COVID-19 condition.

## 2. Materials and Methods

### 2.1. Study Design

A scoping review was performed following the Joanna Briggs Institute’s (JBI) methodology. The findings were reported using the PRISMA-ScR reporting guidelines [11]. The review protocol was registered on the Open Science Framework platform https://osf.io/4x79a/ (access on 23 March 2023).

### 2.2. Research Question

The Population, Concept, and Context (PCC) framework was adopted to formulate the research question [12], as shown in Table 1. The research question was: Which support models exist for caregivers and patients with the post-COVID-19 condition?

### 2.3. Search Strategy and Information Sources

The search terms used in the search strategy (Health Sciences Descriptors [DECS], Medical Subject Headings [MeSH], and Emtree [Embase Subject Headings]) were defined based on the PCC framework. We considered the post-COVID-19 condition as the context, which is also a new theme linked to some controlled vocabularies as a supplementary concept but not yet as MeSH or Emtree terms. A preliminary search was conducted to identify studies and new terms in the titles, abstracts, and subject fields—descriptors, MeSH terms, and keywords. After defining the search strategy, as demonstrated in Table 2 (see PubMed/Medline), we used it in other databases/libraries/portals. The Boolean operators AND and OR combined the sets of terms.

The searches were carried out in May 2022 and updated in October 2022 in the databases and the grey literature. The sources of information were the Virtual Health Library (VHL) under the responsibility of the Latin American and Caribbean Center on Health Sciences Information (BIREME in Portuguese) and its main databases—Latin American and Caribbean Health Sciences Literature (LILACS), Spanish Bibliographic Index of the Health Sciences (IBECS), Nursing Database (BDENF), São Paulo State Health Department, and National Collection of Sources of Information from SUS (ColecionaSUS) among others. The following databases were also searched through the CAPES Portal: PubMed/MEDLINE and PubMed Central (National Library of Medicine), Embase and Scopus (Elsevier), Web of Science (Clarivate Analytics), Cumulative Index to Nursing and Allied Health Literature (CINAHL), Academic Search Premier (APS), American Psychological Association (APA, Ebsco), Scientific Electronic Library Online (Scielo), and Epistemonikos Database. Finally, the grey literature comprised the Science.gov (USA.gov) portal and the websites of the Ministries of Health of 187 countries. The search results were imported into the Endnote reference manager (Version X8.2 Clarivate Analytics, Philadelphia, PA, USA) and, after deduplication, exported to Rayyan QCRI online platform [15].

### 2.4. Eligibility Criteria and Study Selection

This scoping review included studies that answer the research question based on the PCC framework, without restrictions to study design, language, or year of publication (since the disease is novel and all studies are recent). The bibliographic references of the studies included in the synthesis were consulted, as recommended by the JBI [11].

Two independent reviewers blinded for each other’s evaluations selected titles and abstracts. Relevant studies were retrieved in full format, and citation details were imported into the JBI System for Unified Management, Assessment, and Review of Information software [12].

The full texts of the selected studies were re-evaluated by the same two reviewers who analyzed the titles and abstracts independently and in a blinded manner. Any discrepancies between reviewers at each step of the selection process were resolved through discussions with a third reviewer—the lead researcher. The screening was only completed when a proportion of agreement above 75% between the three reviewers was met [12]. The reasons for exclusion were recorded and reported, as shown in Figure 1.

### 2.5. Variables and Strategies to Avoid Bias

Variables of interest included year and country of publication, health policies, models of support, and social support strategies for patients with the post-COVID-19 condition and home caregivers. The following strategies were adopted to avoid the most common types of bias in scoping reviews (selection, information, and confounding bias): (1) the non-use of study design, language, or year restriction, (2) study selection and data extraction conducted by two independent reviewers, with further differences reconciled by consulting a third reviewer, and (3) the adoption of strict inclusion and exclusion criteria to improve homogeneity among primary studies.

### 2.6. Presentation of Results

A map was created showing the regions of the world in which studies were found describing home care and support services for patients with the post-COVID-19 condition and caregivers. Then, a narrative synthesis was created according to the following thematic categories identified by the authors: (1) guidelines and health policies for the post-COVID-19 condition, (2) services and home support for post-COVID-19 condition patients, and (3) support services for home caregivers of patients with the post-COVID-19 condition.

## 3. Results

A total of 3258 records were identified through the electronic search. However, 1245 duplicate titles were excluded resulting in 2013 articles. After the complete screening, 20 articles were included in the final sample (Figure 1). The articles were published between 2020 and 2022 and conducted in 10 countries. Figure 2 demonstrates the mapping of health policies, models of support, and social support strategies aimed at patients with the post-COVID-19 condition and home caregivers.

### 3.1. Guidelines and Health Policies for the Post-COVID-19 Condition

Treating patients with the post-COVID-19 condition is an emerging challenge for rehabilitation health policies, and a new organizational design to provide continued rehabilitation care to this population is necessary [16]. Researchers with different backgrounds, including members of the ISARIC Consortium, the US Centers for Disease Control and Prevention (CDC), experts involved in the WHO post-COVID condition clinical characterization group, leaders of international COVID-19 cohorts, members of the Core Outcome Measures for post-COVID-19 condition/long-COVID initiative, and patient representatives have advocated the urgent development of a Core Outcome Set for research and clinical practice in the post-COVID-19 condition, improving data quality, harmonization, and comparability across different geographic locations [17].

Countries have published clinical guidelines on the post-COVID-19 condition to guide health professionals on the concept, signs, symptoms, and services for treatment, including telerehabilitation [18,19,20,21]. Italian authors published a consensus on treating the post-COVID-19 condition in children [22]. The Brazilian Ministry of Health has published Technical Note No. 133/2021 with recommendations for identifying and following patients with post-COVID-19 conditions at home, using teleconsultation, telemonitoring, and telerehabilitation strategies [18].

The British Thoracic Society [23] adopted an existing pulmonary rehabilitation model in England to care for patients with the post-COVID-19 condition. The model encompasses actions from 6 to 8 weeks, mediated by a trained health team, with telephone follow-up 2 to 3 days after hospital discharge, reference for primary health care, treatment, evaluation, and monitoring of rehabilitation at home, in person, and digitally. An information technology project developed in Italy aims to support Italian regional health systems, offering a model for organizing a network dedicated to people after the acute phase of COVID-19 [16]. This proposal lists several benefits, such as the guarantee of continuity of hospital-outpatient home care, early assessment for clinical diagnosis and treatment of the post-COVID-19 condition, and optimal clinical governance with equal access to rehabilitation care, improving patient and family satisfaction, reducing disabilities, and improving the quality of life. This model may be applicable in other countries, emphasizing communication and teamwork (involving health professionals, health managers, patients, and caregivers) to achieve optimal clinical governance and health care management [16].

### 3.2. Services and Home Support for Post-COVID-19 Condition Patients

Most authors [20,23,24,25,26,27,28,29,30] presented the need for home monitoring of patients with the post-COVID-19 condition by a multi-professional health team and with the adoption of the telerehabilitation strategy. The multidisciplinary team is made up of physicians from different specialties (comorbidities and pharmacological management), nurses (hospital/home care transition, pharmacological management, and basic care during activities of daily living), psychologists (psychosocial support), physiotherapists (pulmonary rehabilitation, fatigue, and mobility), speech therapists (dysphagia), and occupational therapists (vocational rehabilitation and instrumental activities of daily living) [30].

Preliminary results of a study using the COVIDREHAB platform for remote rehabilitation of people with the post-COVID-19 condition in Russia presented an optimization of the physician’s work associated with the prevention of secondary complications and reduction of serious adverse effects [25].

In Ireland, a hybrid model of virtual and face-to-face clinics was established, supported by a multidisciplinary team comprising respiratory services, intensive care, infectious diseases, psychiatry, and psychology. This model identifies patients who need effective follow-up after hospitalization for COVID-19 and aims to support patients with the post-COVID-19 condition and those in need of integrated community care, emphasizing mental health [26].

In Tunisia, an integrated model was developed for cardiorespiratory care of patients with post-COVID-19 at home [31]. The Ministry of Health of Chile reported an online training program carried out by the staff of a tertiary hospital for patients with the post-COVID-19 condition involving the basic aspects of this condition and general recommendations of home rehabilitation to treat symptoms such as general fatigue, fatigue, and pain through physical exercises, nutritional recommendations, voice and swallowing rehabilitation, and mental health interventions [32].

A North American survey highlighted the important role of online communities, through the social networks Facebook and Reddit, to support and exchange information and experiences among patients with the post-COVID-19 condition. Online communities have been useful in understanding the disease and the forms of treatment [33].

A study developed in underserved communities in the United Kingdom assessed the digital accessibility of patients with the post-COVID-19 condition and its acceptability. The results showed that deprived people need strategic community assistance, based on community support organizations, to adhere to intervention programs for the post-COVID-19 condition [24].

### 3.3. Support Services for Home Caregivers of Patients with the Post-COVID-19 Condition

Only one study was found involving support services for home caregivers [10]. The study was developed in Iran and investigated the effectiveness of training for home caregivers of patients with the post-COVID-19 condition regarding four aspects of rehabilitation—musculoskeletal, respiratory, gastrointestinal, and deep vein thrombosis prevention [10]. Other authors only mentioned the need for health professionals’ training of family members/home caregivers in their home environment to assist in rehabilitating patients with the post-COVID-19 condition, especially those with severe sequelae [19,29].

## 4. Discussion

In 2017, the World Health Organization published a plan, “Rehabilitation 2030—A call for action”, highlighting the central role of rehabilitation in public health. As an essential component of acute health care, rehabilitation aims to reduce disability, which allows patients to live in the community and return to their previous level of social participation [34].

The WHO and the Pan American Health Organization (PAHO) emphasize that rehabilitation services where patients reside are always the best places for long-term treatment. In this sense, it is up to managers to prepare community health and psychosocial care services for the increased demands after severe cases of COVID-19, especially among the elderly [35].

Post-COVID-19 functional impairment can impair the ability to perform activities of daily living and functionality, alter work performance, and make social interaction difficult. Furthermore, individuals may become more sedentary, increasing the risk of comorbidities. Health services need to readjust with strategies to provide physical functional recovery and social reintegration of these individuals through rehabilitation [26,29]. In severe COVID-19 patients and those treated in intensive care units (ICU) for a long time, there are also risk factors such as cognitive impairment, acute brain dysfunction, hypoxia, and arterial hypotension, observed in 30–80% of post-ICU patients. At least 25% of these patients become disabled, with a dramatic drop in independence during the years following ICU admission, leaving informal and formal caregivers overwhelmed [25].

Assessing post-COVID-19 patients’ needs also requires a focus on primary health care and patient–family–community rehabilitative care. Financial resources are limited given the numerous healthcare demands, and thus, new strategies, such as multidisciplinary and supervised telerehabilitation programs, represent a good alternative for evaluating and managing post-COVID-19 patients [28]. This recommendation from the Brazilian Unified Health System (SUS) reinforces the need to strengthen Primary Healthcare (PHC) in Brazil through human and material resources for adequate care of post-COVID-19 patients by PHC teams, which occupy a privileged place, as they operate in the territorial logic of healthcare, centered on the patient and constituting a capillary healthcare services network [30,36].

It is important to consider accessing support and referrals when needed. People with the post-COVID-19 condition should not be excluded from the same level of support offered by these tools if they lack digital access or literacy [24]. Mobile health has been increasingly used to guarantee timely medical information delivery. Digital health literacy requires skills complementary to general literacy. Therefore, limited digital literacy may impact some groups negatively, especially deprived populations. Furthermore, populations with limited health literacy face additional challenges with digital health services [37].

Another issue to be discussed is the individual approach presented by the guidelines for patients with the post-COVID-19 condition. Faced with the sequelae presented by patients with this condition, the home caregiver experiences a physical and emotional overload in the face of a necessary readaptation process in the entire environment surrounding the patient. Studies with family caregivers of people with chronic and disabling diseases highlight that the sphere of priorities is established around the most disabled family member, and the needs of other family members are less evident, causing changes in the roles assumed by each one [38,39,40]. All these changes experienced by the family caregiver can negatively influence their emotional state, leading to high levels of psychological distress [9,39,41,42].

The study developed in Iran showed how crucial the education of caregivers about rehabilitation measures is, showing the level of knowledge of caregivers by the application of a questionnaire before the educational intervention and then the improvement of home care after the application of educational pamphlets. It noted that the level of knowledge of caregivers about the rehabilitation care provided to patients with COVID-19 was not satisfactory [10].

A study in two regions of Peru showed that the recovery process of post-COVID-19 patients largely depends on the work of the caregivers. Therefore, caregivers’ self-care and quality of life must be promoted, including physical, spiritual, and psychological well-being. The results of the study also led the authors to the conclusion that there is an association between depression and poor quality of life in caregivers, and it was found that having children, receiving the vaccine, and being a direct relative of the person receiving care (which implies that the caregiver is a very close family member) were associated with a good quality of life [9].

It is necessary to monitor caregivers’ mental health and develop strategies to adapt to pandemic conditions. It would be advisable to allocate caregivers fewer working hours and greater inclusion in social activities, reflecting better patient care and faster recovery [9]. Thus, there is an urgent need to develop and implement home social support and support services aimed at patients with the post-COVID-19 condition and their home caregivers, both in terms of training and meeting the needs of care presented by patients and caregivers.

### Strengths and Limitations

The main strength of this study is that it was based on a rigorous process that included several databases and the grey literature, without restrictions on study design, language, or year of publication, allowing us to identify studies from different regions of the world. We were able to present a synthesis of existing support models for caregivers of patients with the post-COVID-19 condition, a synthesis not yet available in the literature. As limitations, the lack of a MeSH term focused on the post-COVID-19 condition is worth noting, which interfered with the search strategy. Furthermore, the fact that the post-COVID-19 condition is not referenced in the traditional academic citation databases or grey literature does not mean that other countries do not have social support models for patients with this condition and their caregivers.

## 5. Conclusions

This study has summarized clinical guidelines, support services, and social support strategies for caregivers and patients with the post-COVID-19 condition. These strategies still have an individual/patient focus and emphasize adopting remote or hybrid home care and training models. Despite the importance of hospital-centered approaches and telerehabilitation (for training and monitoring the patient–family dyad), barriers such as lack of access and digital literacy must be considered, especially for people with socioeconomic disadvantages. Moreover, is noteworthy that only 10 countries reported in the studies had reference services for the home care of patients with the post-COVID-19 condition.

Given the above, we recommend that researchers, practitioners, and policymakers update existing guidelines for the home treatment of post-COVID-19 conditions in a clear and consistent manner considering the socioeconomic and cultural barriers patients and caregivers may face.

## Figures and Tables

**Figure 1 ijerph-20-02563-f001:**
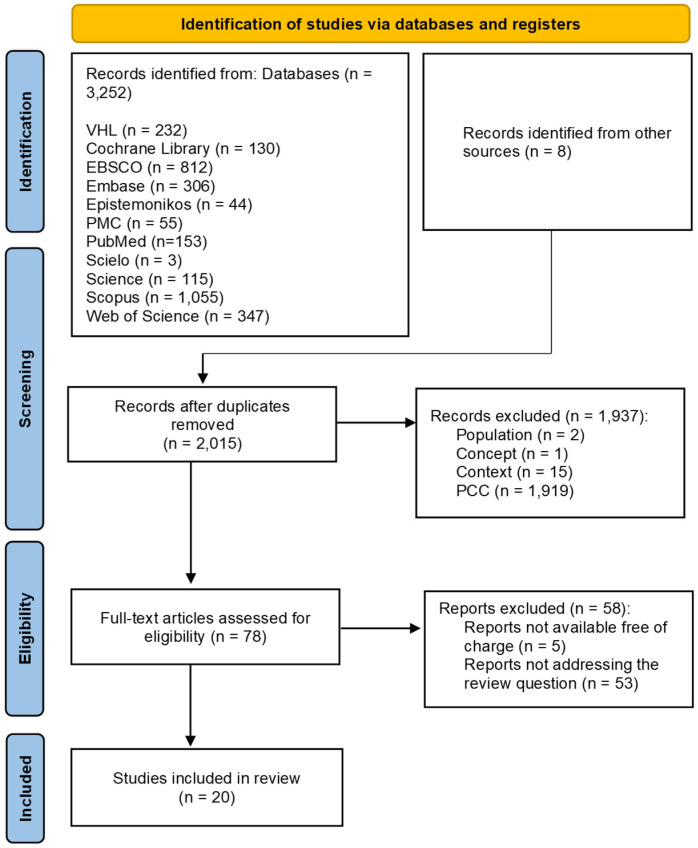
PRISMA-ScR flow diagram.

**Figure 2 ijerph-20-02563-f002:**
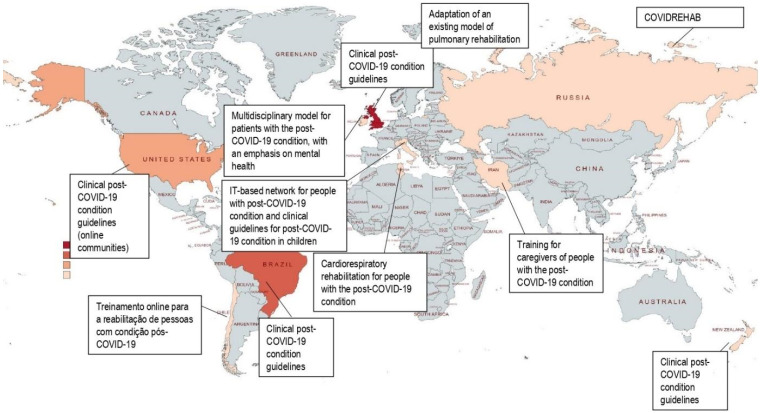
Health policies and models of social support for caregivers and patients with the post-COVID-19 condition.

**Table 1 ijerph-20-02563-t001:** PCC framework for defining the eligibility of the studies.

PCC	Definitions	
Population	Patients with the post-COVID-19 condition	People with neuropsychiatric, motor, or cardiopulmonary sequelae due to COVID-19 infection and who, from this, have some functional dependence to perform their daily activities [1].
	Home caregiver	Someone who takes care of the objectives established by a specialized institution or a directly responsible person. Their function ensures the assisted person’s well-being, health, food, personal hygiene, education, culture, recreation, and leisure. The person in the family or community provides care to another person of any age who requires care due to physical or mental limitations, with or without remuneration [13].
Concept	Models of organization of support and home support	Structured services include specific care actions for people, families, and caregivers to offer adequate conditions for care to preserve family and social life [13].
Context	Post-COVID-19 Condition	Illness that occurs in individuals with a history of probable or confirmed infection with SARS-CoV-2, usually 3 months after the onset of COVID-19, with symptoms that last for at least 2 months and cannot be explained by an alternative diagnosis. Common symptoms include fatigue, shortness of breath, and cognitive dysfunction [14].

**Table 2 ijerph-20-02563-t002:** Search strategy conducted on the PubMed Portal.

Search	Query	Results
#1	“Caregivers”[mh] OR Caregiver*[tiab] OR Carer**[tiab] OR “Care Givers”[tiab] OR “Care Giver”[tiab] OR family care*[tiab] OR “unpaid care”[tiab] OR informal care*[tiab] OR “Family”[mh] OR Families[tiab] OR Filiation[tiab] OR relatives[tiab] OR Stepfamil*[tiab] OR Parent*[tiab] OR “Step Parents”[tiab] OR Step-Parent[tiab] OR Step-Parent*[tiab] OR Stepparent*[tiab] OR maternity[tiab] OR motherhood[tiab] OR parenthood[tiab] OR paternity[tiab] OR “mothers”[mh] OR “Fathers”[mh] OR “mothers”[tiab] OR ”Fathers”[tiab] Sort by: Most Recent	1,128,550
#2	“Social Support”[mh] OR “Social Support”[tiab] OR “Social Care”[tiab] OR “Psychosocial Support Systems”[tiab] OR “Psychological Support System”[tiab] OR “Psychological Support Systems”[tiab] OR “Psychosocial Support”[tiab] OR “Psychosocial Support System”[tiab] OR “Psychosocial Supports”[tiab] OR “Social Support System”[tiab] OR “Social Support Systems”[tiab] OR “Home Nursing”[mh] OR “home nursing”[tiab] OR “Non-Professional Home Care”[tiab] OR “Nonprofessional Home Care”[tiab] OR “Home Care Services”[tiab] OR “Domiciliary Care”[tiab] OR “Domiciliary Care”[tiab]”Home Care”[tiab] OR “Home Care Service” OR “Home Health Care” OR “Home Care Services, Hospital-Based”[mh] OR “Hospital Based Home Care”[tiab] OR “Hospital Based Home Care Services”[tiab] OR “Hospital Based Home Cares”[tiab] OR “Hospital Home Care Services”[tiab] OR “Hospital-Based Home Care”[tiab] OR “Hospital-Based Home Care Services”[tiab] OR “Hospital-Based Home Cares”[tiab] OR “Orientation”[mh] OR Orientation*[tiab] OR support[tiab] OR “Patient Care”[mh] OR “Informal care”[tiab] OR “Informal cares”[tiab] OR “Health Services”[mh] OR “Health Service”[tiab] OR “Rehabilitation”[mh] OR “Rehabilitation Centers”[mh] OR Rehabilitation Center*[tiab] OR Long-term care facilities[tiab] OR “Long term care”[tiab] OR “Long-Term Care”[mh] Sort by: Most Recent	3,730,959
#3	((COVID-19[mh] OR “COVID 19”[tiab] OR “SARS-CoV-2”[tiab] OR “Novel Coronavirus”[tiab] OR 2019-nCoV[tiab] OR “Coronavirus Disease 2019”[tiab] OR “Coronavirus Disease-19”[tiab] OR “Coronavirus Disease 19”[tiab] OR “Severe Acute Respiratory Syndrome Coronavirus 2 Infection”[tiab] OR “SARS Coronavirus 2 Infection”[tiab] OR “2019 nCoV”[tiab] OR COVID19[tiab] OR SARS-CoV-2[tiab] OR “SARS Coronavirus 2 Infection”[tiab] OR “Wuhan Coronavirus”[tiab] OR SARS-CoV-2[tiab]) AND (complication[tiab] OR “associated conditions”[tiab] OR “associated disease”[tiab] OR “coexistent conditions”[tiab] OR “coexistent disease”[tiab] OR “concomitant conditions”[tiab] OR “concomitant disease”[tiab] OR sequelae[tiab] OR sequel*[tiab])) OR “COVID-19 sequalae”[tiab] OR “Core Outcome Set”[tiab] OR “Long COVID”[tiab] OR Post-COVID-19[tiab] OR “Post-acute sequelae of SARS-CoV-2 infection”[tiab] OR “Post COVID 19”[tiab] OR “chronic COVID syndrome”[tiab] OR “chronic COVID-19”[tiab] OR “COVID long-hauler”[tiab] OR “long haul COVID”[tiab] OR “long hauler COVID”[tiab] OR “post COVID 19 fatigue”[tiab] OR “post COVID 19 neurological syndrome”[tiab] OR “post COVID 19 syndrome”[tiab] OR “post COVID fatigue”[tiab] OR “post COVID syndrome”[tiab] OR “post-acute COVID syndrome”[tiab] OR “post-acute COVID-19”[tiab] OR “post-acute COVID-19 syndrome”[nm] OR “post-acute COVID-19 syndrome”[tiab] OR long-COVID [tiab] OR “long-haul COVID”[tiab] OR “persistent COVID-19”[tiab] OR “post-acute COVID19 syndrome”[tiab] Sort by: Most Recent	10,299
#4	#1 AND #2 AND #3 Sort by: Most Recent	142

## Data Availability

The data presented in this study are available in the article.

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
