# Peer review of "Models of Support for Caregivers and Patients with the Post-COVID-19 Condition: A Scoping Review"

_ijerph, 2023, doi:10.3390/ijerph20032563_

Round 1
Reviewer 1 Report
Models of support for caregivers and patients with the post-COVID-19 condition: a scoping review.
1. Title: it would be useful to write it using more keywords to improve its reachability. E.g., Post-Acute COVID-19 Syndrome (MeSH ID: D000094024) instead of “post-COVID-19 condition”.
2. Abstract: label the parts of its structure, include a background for the disease.
3. Introduction:
- Briefly define what is COVID-19 and mention some of its consequences before explaining what the post-COVID-19 condition is.
- Mention which “knowledge gaps” will this review fill in an explicit statement.
4. Methods:
- State the version for EndNote and Rayyan that were used.
- Mention that the Ministries of Health of 187 countries were included as grey literature, not as part of the main search strategy.
- Explain why there is not a limit to the year of publication.
- List all variables for which data were sought before the “Results” section.
- Identify any way of addressing possible bias of the included studies.
5. Format: Ensure that font size and type are according to the style. E.g., table 2 font size is significantly bigger than the rest of the text.
Author Response
Thank you for your comments. We have carefully reviewed the comments and have revised the manuscript accordingly. Our responses are given in a point-by-point manner. Changes in the manuscript are shown in yellow.
- Title: it would be useful to write it using more keywords to improve its reachability. E.g., Post-Acute COVID-19 Syndrome (MeSH ID: D000094024) instead of “post-COVID-19 condition”.
Response: We appreciate your suggestion, but the term “post-COVID-19 condition” used in the title is the same term used by the Centers for Disease Control and Prevention (CDC) to characterize the wide range of physical and mental health consequences experienced by some patients four or more weeks after SARS-CoV-2 infection, which is the population focused on our scoping review. Therefore, we decided to keep the term in the title and consistently in the manuscript. For additional clarifications, please visit https://www.cdc.gov/coronavirus/2019-ncov/hcp/clinical-care/post-covid-conditions.html.
- Abstract: label the parts of its structure, include a background for the disease.
Response: The suggestions were incorporated in the abstract.
- Introduction:
- Briefly define what is COVID-19 and mention some of its consequences before explaining what the post-COVID-19 condition is.
- Mention which “knowledge gaps” will this review fill in an explicit statement.
Response: The suggestions were incorporated in the Introduction.
- Methods:
- State the version for EndNote and Rayyan that were used.
- Mention that the Ministries of Health of 187 countries were included as grey literature, not as part of the main search strategy.
- Explain why there is not a limit to the year of publication.
Response: The suggestions above were incorporated in the Introduction.
- List all variables for which data were sought before the “Results” section.
- Identify any way of addressing possible bias of the included studies.
Response: Thank you for these important suggestions. We have added a subsection called “2.4. Variable and strategies to avoid bias” into the Methods section and incorporated the information requested in the comments above.
- Format: Ensure that font size and type are according to the style. E.g., table 2 font size is significantly bigger than the rest of the text.
Response: We have incorporated the recommended changes in the manuscript.

Reviewer 2 Report
This manuscript is timely, relevant, well-written, and adds greatly to the knowledge on this developing topic. Post-Covid-19 is recognized, but little is still known, especially in the area of caregiver support. The literature review was focused, succinct, and led appropriately to the identified gap of the lack of support models for caregivers. One area that could have been explored is to look at other models of support for other chronic conditions requiring ongoing rehabilitation (such as cardiac), and explore if these support models could be adapted to the needs for those with post-Covid-19.
The design of the scoping review was explained well. The search strategy (Fig 1) was excellent and displayed well in diagrammatic form, as was the global scope of the findings in 10 countries (Fig 2). The results were identified and explained well, and the linking to the countries added the understanding of the global reach of this developing problem.
This study provides a basic understanding of the current status of caregiver support for post-Covid-19 patients (not much there), and provides an impetus to develop these support models further. Excellent work!
Author Response
Thank you for your comments. We have carefully reviewed the comments and have revised the manuscript accordingly. Our responses are given in a point-by-point manner. Changes in the manuscript are shown in yellow.
Review 2:
- This manuscript is timely, relevant, well-written, and adds greatly to the knowledge on this developing topic. Post-Covid-19 is recognized, but little is still known, especially in the area of caregiver support. The literature review was focused, succinct, and led appropriately to the identified gap of the lack of support models for caregivers. One area that could have been explored is to look at other models of support for other chronic conditions requiring ongoing rehabilitation (such as cardiac), and explore if these support models could be adapted to the needs for those with post-Covid-19.
- The design of the scoping review was explained well. The search strategy (Fig 1) was excellent and displayed well in diagrammatic form, as was the global scope of the findings in 10 countries (Fig 2). The results were identified and explained well, and the linking to the countries added the understanding of the global reach of this developing problem.
- This study provides a basic understanding of the current status of caregiver support for post-Covid-19 patients (not much there), and provides an impetus to develop these support models further. Excellent work!
Response: Thank you so much for taking the time to give us your feedback. We appreciate your comments and reviews! Concerning the comment about other models of support for other chronic conditions requiring rehabilitation, this is an interesting issue and we have plans to investigate it in future studies.

Reviewer 3 Report
Thank you for the opportunity to review this interesting manuscript. The authors conducted a scoping review for the models of support for caregivers and patients with the post- 2 COVID-19 condition. The authors are to be congratulated on their findings and achievements on this issue.
In my opinion, the authors provided an interesting and sound report. The objectives were clearly stated. The study method was adequately described. The results clearly presented. The discussion pointed out the important findings. The conclusions appropriately based on the results and discussions.
I am appreciated for the fact and congratulated on authors’ achievement. however, some concerns had raised from this work. The superscript “***” was not found in table 1. I am confused by the numbers in Figure 1 PRISMA-ScR flow diagram. I am wondering why the box showed n=78 then n=76 in the next box after records excluded (n=1937). Furthermore, I am appreciated if the authors might revise the figure 1 to present an appropriate way for the box of duplicates. I am appreciated if the authors might provide the strength and limitations in the discussion section. Line 307-311 might be discussed in the limitation section.
Author Response
Thank you for your comments. We have carefully reviewed the comments and have revised the manuscript accordingly. Our responses are given in a point-by-point manner. Changes in the manuscript are shown in yellow.
- Thank you for the opportunity to review this interesting manuscript. The authors conducted a scoping review for the models of support for caregivers and patients with the post- 2 COVID-19 condition. The authors are to be congratulated on their findings and achievements on this issue.
- In my opinion, the authors provided an interesting and sound report. The objectives were clearly stated. The study method was adequately described. The results clearly presented. The discussion pointed out the important findings. The conclusions appropriately based on the results and discussions.
- I am appreciated for the fact and congratulated on authors’ achievement.However, some concerns had raised from this work.
Response: Thank you so much for taking the time to give us your feedback!
- The superscript “***” was not found in table 1. I am confused by the numbers in Figure 1 PRISMA-ScR flow diagram. I am wondering why the box showed n=78 then n=76 in the next box after records excluded (n=1937). Furthermore, I am appreciated if the authors might revise the figure 1 to present an appropriate way for the box of duplicates.
Response: Thank you for your suggestion. We decided to redraft Figure 1. We incorporated the information that was signaled by the superscript * into the first box of the figure. We made other changes and believe that the figure looks better and is clearer (please observe that the number of included and excluded records at each stage are now correct).
- I am appreciated if the authors might provide the strength and limitations in the discussion section. Line 307-311 might be discussed in the limitation section.
Response: Thanks for your suggestion! We have added subsection “4.1 Strengths and limitations” into the Discussion and believe that this was an important addition to the paper.

Reviewer 4 Report
Your study question is an excellent one and you have done a good job of answering the questions, O have only three point that need to be clarified.
1. & 2; On lines 125 and 125 you discuss two independent reviewers. It is not clear, from the way you have presented this, whether there are a total of 5 reviewers or whether 2 reviewers are used twice, with a tie-breaker given by either the third or the fifth reviewer. Please clarify.
3. line 292: what do you mean by the phrase "being a direct relative" mean? Does it mean living near family, or having a large family, or being a caregiver to a family member? Or something else? Please clarify.
Author Response
Thank you for your comments. We have carefully reviewed the comments and have revised the manuscript accordingly. Our responses are given in a point-by-point manner. Changes in the manuscript are shown in yellow.
- Your study question is an excellent one and you have done a good job of answering the questions, I have only three points that need to be clarified.
- & 2; On lines 125 and 125 you discuss two independent reviewers. It is not clear, from the way you have presented this, whether there are a total of 5 reviewers or whether 2 reviewers are used twice, with a tie-breaker given by either the third or the fifth reviewer. Please clarify.
Response: There were 2 reviewers and discrepancies were resolved by a third reviewer. We have made changes in the manuscript to clarify this information (see the sentences in yellow highlight).
- line 292: what do you mean by the phrase "being a direct relative" mean? Does it mean living near family, or having a large family, or being a caregiver to a family member? Or something else? Please clarify.
Response: Thanks for your suggestion. We added information in the text to clarify this point.

Reviewer 5 Report
Thank you for the opportunity to review the manuscript. It is well done and written.
Following the guidelines of the PRISMA Scoping Review Guidelines I recommend creating a chart with the 20 studies that were selected for review to include author, article type, population, and findings identified (perhaps country of origin can be included as well). The graph of the graph shown on Fig 2, although helpful, it only shows 11 lit sources worldwide.
Are there any recommendations for future research? and to whom would the recommendations be directed at?
Author Response
Thank you for your comments. We have carefully reviewed the comments and have revised the manuscript accordingly. Our responses are given in a point-by-point manner. Changes in the manuscript are shown in yellow.
- Thank you for the opportunity to review the manuscript. It is well done and written.
- Following the guidelines of the PRISMA Scoping Review Guidelines I recommend creating a chart with the 20 studies that were selected for review to include author, article type, population, and findings identified (perhaps country of origin can be included as well). The graph shown on Fig 2, although helpful, it only shows 11 lit sources worldwide.
Response: Thank you for this relevant suggestion. However, we decided not to create the chart because most of the information that would be included in it is already described in the article textually and in Figure 2 (specifically country and caregiver/patient support model used). As for the year of publication, all studies are recent because the disease is novel, so we believe it is of little relevance to present this type of data in a chart. Finally, note that the sample of publications includes not only scientific articles but also guidelines, and not all information applicable to a scientific article is present in this type of document.
- Are there any recommendations for future research? and to whom would the recommendations be directed at?
Response: Thank you for asking these important questions. We have made changes in the Conclusion section based on your questions and believe that these changes were an excellent addition to our manuscript.

Round 2
Reviewer 1 Report
Thanks for the corrections and improvements to the manuscript. Congratulations to authors.